# Activating the DNA Damage Response and Suppressing Innate Immunity: Human Papillomaviruses Walk the Line

**DOI:** 10.3390/pathogens9060467

**Published:** 2020-06-13

**Authors:** Claire D. James, Dipon Das, Molly L. Bristol, Iain M. Morgan

**Affiliations:** 1Philips Institute for Oral Health Research, School of Dentistry, Virginia Commonwealth University (VCU), Richmond, VA 23298, USA; cdjames@vcu.edu (C.D.J.); ddas@vcu.edu (D.D.); mlbristol@vcu.edu (M.L.B.); 2VCU Massey Cancer Center, Richmond, VA 23298, USA

**Keywords:** human papillomavirus, cervical cancer, head and neck cancer, DNA damage response, innate immune response, interaction

## Abstract

Activation of the DNA damage response (DDR) by external agents can result in DNA fragments entering the cytoplasm and activating innate immune signaling pathways, including the stimulator of interferon genes (STING) pathway. The consequences of this activation can result in alterations in the cell cycle including the induction of cellular senescence, as well as boost the adaptive immune response following interferon production. Human papillomaviruses (HPV) are the causative agents in a host of human cancers including cervical and oropharyngeal; HPV are responsible for around 5% of all cancers. During infection, HPV replication activates the DDR in order to promote the viral life cycle. A striking feature of HPV-infected cells is their ability to continue to proliferate in the presence of an active DDR. Simultaneously, HPV suppress the innate immune response using a number of different mechanisms. The activation of the DDR and suppression of the innate immune response are essential for the progression of the viral life cycle. Here, we describe the mechanisms HPV use to turn on the DDR, while simultaneously suppressing the innate immune response. Pushing HPV from this fine line and tipping the balance towards activation of the innate immune response would be therapeutically beneficial.

## 1. Introduction

Human papillomaviruses activate the DNA damage response and suppress innate immunity, two processes that are required for a successful viral life cycle [1,2,3,4,5,6]. There is a link between these two pathways in non-HPV cells; DNA damage can induce the innate immune response, resulting in interferon production and potential attenuation of cellular proliferation [7,8,9,10,11,12]. Figure 1 summarizes how HPV can activate the DDR, including via replication stress on the replicating viral DNA. This replication stress could result in the production of DNA fragments that could egress to the cytoplasm and activate the innate immune response (Figure 2). To combat this, HPV have multiple mechanisms for repressing the innate immune response (Figure 3). The purpose of this review is to reflect on how HPV manipulate these two pathways to allow progression of the viral life cycle, and to highlight some known interactions between the two pathways regulated by HPV. Inhibition of the DDR blocks the HPV life cycle and is a strategy for controlling infection and treating cancers. Similarly, activating innate immunity, while not disrupting HPV activation of the DDR, could be a useful therapeutic tool for combating HPV disease. We will begin the review by describing the activation and manipulation of the DDR by HPV. How HPV regulate the innate immune response will follow. Finally, we will highlight interactions between the DDR and innate immunity in HPV cells mediated by the viral replication complex.

## 2. The DDR during the HPV Life Cycle

Papillomaviruses are divided into low-risk (LR) and high-risk (HR) types, with the criteria for HR-HPV types being that they cause cancer [13]. This review focuses on HR-HPV, and henceforth “HPV” refers to HR-HPV. HPV infect basal cells of the epithelium via abrasions that allow access for viral particles [6,14,15]. Following infection, HPV must wait for mitosis in order for the virus to enter the nucleus and begin the viral life cycle [16,17]. Once located to the nucleus, transcription occurs from the viral long control region (LCR), mediated by host cellular factors [18]. This results in a transcript processed into RNA species that are translated to produce the viral proteins. The HPV oncoproteins E6 and E7 target a host of cellular proteins, including p53 and pRb, respectively. These oncoproteins promote proliferation of the infected cell and protect it from growth attenuation, therefore promoting the viral life cycle [19,20,21,22,23]. Two viral proteins, E1 and E2, are required for replication of the viral genome in association with host proteins [24,25,26]. E2 is a DNA binding factor that forms homodimers and binds to specific 12 bp palindromic sequences in the LCR that surround an A/T-rich origin of replication [27]. Following binding to its target sequences, E2 recruits the viral helicase E1 that forms a di-hexameric complex responsible for replicating the viral genome in association with host polymerases [28,29]. Following the initiation of replication in the infected cell, the viral genome copy number establishes at multiple copies per cell. As the proliferating infected cell moves through the epithelium, the copy number is maintained. In the upper layers of the epithelium, an amplification stage increases the viral genome copy number in a population of the differentiated cells. At this stage of the viral life cycle, the viral structural proteins L1 and L2 encapsulate the viral genomes, forming viral particles. These viral particles then egress from the top layer of the differentiated epithelium.

During the establishment phase of the viral life cycle, the amplification of the viral genome from one to multiple copies per cell likely activates the DDR due to torsional stress on the viral genome caused by clashing replication forks [30]. Three viral proteins can activate the DDR when expressed in isolation: E1, E6 and E7. E1 binds cellular DNA indiscriminately, activating DNA unwinding and replication via its helicase activity and interaction with cellular polymerases [31,32,33,34,35]. E1, and its binding partner E2, require interactions with host DNA replication factors and DDR proteins to carry out viral replication [36,37,38,39]. The E1 expression alone induces various DNA damage markers, likely due to its helicase activity; moreover, E1-E2-containing viral replication foci co-localize with various DDR markers and confer replication fidelity [40,41].

E6 targets p53 for degradation, therefore altering the cellular response to DNA damage; p53 is integral to a healthy response to DNA damage in order to promote cell cycle arrest followed by DNA repair, or apoptosis to kill the damaged cell [21]. E7 disrupts the function of the tumor suppressor pRb in order to promote the transition from the G1 to S phase [19]. In addition, E7 disrupts several processes during mitosis that can result in genomic instability and the combined functions of E7 stimulate activation of the DDR [42,43,44,45]. These studies employed overexpression of viral proteins, and monitored their effect on the cell and the DDR pathway. We recently reported that HPV16 activates the DDR in human keratinocytes irrespective of the expression of E6 and E7 (Figure 4) [46]. This was done by introducing stop codons into the E6 and E7 genes (individually and combined) in the full HPV16 genome and generating N/Tert-1 (human foreskin keratinocytes immortalized by telomerase) cells containing wild type and mutant genomes. N/Tert-1 cells support late stage markers of the HPV16 life cycle and are transcriptionally reprogrammed by the virus [47,48]. Strikingly, the DDR activation and the upregulation of replication stress response genes were identical in the N/Tert-1 cells with HPV16 genomes irrespective of mutations in E6 or E7, or in both. This demonstrates that viral replication per se likely directly activates the DDR and this is logical based on the challenge viral replication poses to the cells when establishing 20–50 copies per cell during the initial infection. Clearly, activation of the DDR by viral replication has the potential to attenuate cellular growth and this is what we observed in our N/Tert-1 system; N/Tert-1 cells expressing the HPV16 genome that lacked the expression of E6 and E7 grew significantly slower than those containing the intact HPV16 genome. Therefore, the viral oncogenes E6 and E7 overcome the DDR originating from viral genome replication that suppresses cell growth in order to promote proliferation. This is supported by the observation that in the presence of the E6 and E7 oncogenes, the DNA damage response is manipulated and host DNA repair factors have an aberrant interaction with the host chromatin [49]. This potentially prevents the host genome “seeing” the DDR signals sent from viral replication. This is likely why, when overexpressed by themselves, E6 and E7 activate the DDR; their manipulation of DNA damage and repair factors away from the host genome would provide replication stress that would activate the DDR. A number of DNA damage and repair proteins interact with replicating viral DNA and promote homologous recombination (HR) repair on the replicating viral DNA [39,41,50,51,52,53,54,55,56,57]. This would repair stalled and broken replication forks and allow high-fidelity replication of the viral genome.

In 2009, Moody and Laimins published a seminal paper demonstrating activation of the DDR by HPV31 in keratinocytes, and that this activation is required for the amplification stage of the viral life cycle [58]. The addition of ATM inhibitors blocked amplification of the viral genome following differentiation of HPV31 containing human keratinocytes. ATM inhibition did not affect maintenance of the viral genomes in monolayer cells. This report also demonstrated that overexpression of E7 by itself can induce the DDR and that E7 and ATM form a cellular complex. Overexpression of E7 can induce increased expression of several proteins involved in DNA replication and repair [59]. Many of the genes encoding these proteins are targets of E2F1, a protein regulated by pRb, a primary target of E7. Subsequent studies demonstrated that a number of host DDR proteins are recruited to HPV replication foci, presumably to facilitate HR-mediated viral DNA replication [41,52,53,54,55,56]. The knock down of several of these factors results in abrogation of the viral genome amplification; in particular, the MRN complex (MRE11-RAD50-NBS1) is crucial in this regard [53]. The MRN complex recognizes double-stranded DNA breaks and is required for efficient HR in human cells. Indeed, BRCA1 and RAD51, two proteins required for processing DNA for efficient HR, also are required for amplification of HPV DNA during differentiation of human keratinocytes [52]. 

Completion of the viral life cycle requires a complex interaction between HPV and the DDR. This interaction allows the proliferation of infected cells with an active DDR, something that ordinarily should not occur. External DNA-damaging agents cause cell cycle arrest, a pause that allows for recognition of the damage, its repair and subsequent re-entry into the cell cycle [60]. Therefore, the DDR turned on by HPV, which initiates internally in the cell, represents a unique activation of the DDR pathway that is a therapeutic target in HPV-positive cells. 

## 3. HPV and Repression of the Innate Immune Response (IIR)

Viral infection is met with a host of defense mechanisms in mammalian cells, and these can target HPV infection. Before discussing the evasion strategies HPV use against these host defense mechanisms, we will briefly introduce the innate immune response. Mammalian cells express cytoplasmic proteins that act as pathogen recognition receptors (PRRs), responsible for recognizing external agents such as RNA and DNA viruses [61]. PRRs then activate a host of cellular response pathways. External DNA-damaging agents can also activate the innate immune response in treated cells, due to fragments of damaged DNA escaping from the nucleus into the cytoplasm [12]. A major pathway activated by recognition of cytoplasmic DNA is the stimulator of interferon genes (STING) pathway [62]. cGAS is the cytoplasmic sensor in the STING pathway; activation of cGAS results in the production of the second messenger GMP-AMP (cGAMP) [63]. cGAMP then interacts with the STING protein on the endoplasmic reticulum, promoting its translocation to the Golgi. This results in recruitment of TANK-binding kinase 1 (TBK1) and IκB kinase (IKK) which phosphorylate interferon regulatory factor 3 (IRF3) and the NF-κB inhibitor IκBα, respectively [64]. IRF3 then dimerizes and enters the nucleus, resulting in transcriptional activation of genes encoding type 1 interferons. IκBα phosphorylation results in NF-κB nuclear entry and activation of the transcription of pro-inflammatory cytokines including IL6 and TNF. Overall, PRRs activate innate immunity, which in turn assists with the activation of adaptive immunity.

HPV have evolved multiple mechanisms for suppressing the IIR. This is likely because HPV genomes are relatively small DNA molecules replicating more than once per cell cycle and could produce DNA fragments that could egress to the cytoplasm and activate the IIR. The consequences of not repressing the IIR, would be a downregulation of cellular proliferation or the induction of senescence that would disrupt the viral life cycle. Lau and colleagues demonstrated that HPV E7 binds to STING and prevents its signaling, therefore preventing the production of interferon [65]. Moreover, E6 can complex with IRF3 and inhibit its transcription activity which would also compromise the STING pathway [66]. Much work remains in order to enhance our understanding of how HPV regulate STING during the viral life cycle. 

Interferon (class 1) activates the transcription of interferon-stimulated genes (ISGs) via a transcription complex called interferon stimulated gene factor 3 (ISGF3) [67,68,69]. This complex consists of STAT1 and STAT2, phosphorylated via interferon signaling by Janus kinases (JAK), that then complex with IRF9 and enter the nucleus to bind specific DNA sequences and activate the transcription of ISGs [70,71]. Following viral infection, there is residual unphosphorylated ISGF3 (UISGF3) activity that results in a longer-term activation of a sub-set of ISGs, thought to act as an innate “memory” that can deal quickly with a subsequent viral infection [72]. Interestingly, long-term exposure of cells to IFN-β results in elevated levels of these ISGs, but also results in resistance to DNA-damaging agents as well as to viruses, providing a further link between the DDR and IIR [73]. Given the anti-viral nature of ISGs, HPV have multiple mechanisms for blocking the JAK-STAT pathway to prevent their activation of ISGs. Four viral proteins, when overexpressed, can repress ISG expression: E6, E7, E1 and E2 [20,47,74,75].

E6 and E7 block interferon signaling at precise points in the pathway. E6 binds to Tyk2 to impair Jak-STAT activation by IFNα, and also promotes resistance to interferon-induced growth arrest via inhibition of p53 acetylation [76,77]. This repression of the innate immune system by HPV promotes the viral life cycle via the subversion of host immunity, but downregulation of STAT-1 is also required for amplification of the viral genome during the HPV life cycle, and there is also a role for STAT5 in this process [78,79,80].

What is the main viral protein that represses the innate immune system, and does this change during the viral life cycle? We began to address this question using our N/Tert-1 system that contained wild type and E6/E7 mutant genomes. To our surprise, elimination of the expression of either oncogene completely relieved the repression of innate immune genes by HPV16 [46]. Therefore, even though overexpression of E6 or E7 by themselves in N/Tert-1 cells represses the expression of innate immune genes, both oncoproteins are required for this repression in the context of the entire HPV16 genome. One notable feature of the cells containing HPV16 genomes with stop codons in both E6 and E7 was that they grew more slowly than cells containing the wild type HPV16 genome. Furthermore, the expression of IFNκ was elevated when compared with control and other HPV16 genome-containing cells. The mechanism for this increase in IFNκ production is unclear, and differences in STING signaling in the E6 and E7 mutant genome cells were not easily detectable. Therefore, E6 and E7 relieve growth suppression induced by E1-E2 replication stress; IIR suppression may contribute to this process. In addition, the cellular protein IFIT1, which is repressed by HPV at the transcription level, can bind directly to HPV18 E1 and has the potential to attenuate HPV18 DNA replication [81,82]. Thus, downregulation of the IFIT1 expression is likely required for the HPV18 life cycle. A more recent study has also clearly demonstrated an essential role for E5 in the repression of innate immunity during the HPV16 life cycle, adding to the multifaceted story of how HPV16 represses innate immunity [83]. Overall, there are five viral proteins that can repress innate immunity (E6, E7, E1, E2 and E5), i.e., almost all the viral proteins expressed during the initial phase of infection. While this is complex, the good news from a therapeutic perspective is that targeting the E5, E6 or E7 mechanism may be sufficient to reactivate the innate immune response in HPV16 containing keratinocytes. 

## 4. Crosstalk between HPV DNA Replication, the DDR, and Innate Immunity

Treatment of cells with drugs such as etoposide results in activation of the IIR, presumably due to disrupted replication producing small DNA fragments that enter the cytoplasm and are bound by PRRs [12]. Multiple viral proteins regulate the DDR (E6, E7, E1, E2 in association with E1) and all of these repress innate immune gene expression along with E5. Much work remains to resolve this complicated network of interactions between innate immunity and the DDR in the context of the HPV16 life cycle. There is a further viral process that crosstalks between the DDR and IIR, and that is viral DNA replication. 

In an HPV-infected cell, high-fidelity host and viral replication are essential to block the release of DNA into the cytoplasm that could trigger the IIR. One way that the virus promotes fidelity of its own replication is to recruit DNA repair factors to the viral genome, where they facilitate homologous recombination (HR)-mediated DNA replication [41,50,52,53]. The MRN complex is recruited to the viral genome. This complex consists of MRE11, RAD50 and NBS1, and following double strand DNA breaks (DSBs), this complex recognizes the DNA damage and binds to the DNA [84]. Following binding to the DSB, the MRN complex slides 100–200 bp away from the broken end. MRE11 then performs an endonuclease cut in the DNA on the strand that has the 5’ free end on the DSB, assisted by another cellular protein, CtIP [85]. The MRN complex then moves back towards the broken edge where MRE11 3’ to 5’ exonuclease activity digests the nucleotides, leaving a 3’ overhang strand on the strand not cut by MRE11 [86]. The MRN complex then detaches from the DSB. Additional exonucleases (such as Exo1 and DNA2) extend the 3’ overhang on the DSB DNA [87,88]. This overhang is around 1000 bp long and is protected by Replication Protein A (RPA) [89]. BRCA2-RAD51 then displace RPA forming a RAD51 ssDNA nucleofilament required for HR [90]. The RAD51 bound DNA then promotes strand invasion, ultimately resulting in HR repair of the broken DNA [91]. The recruitment of the MRN complex promotes activation of the DDR as NBS1 complexes with ATM and activates its activity following MRN binding to DSB DNA [92]. 

A striking feature of the MRN complex and RAD51 is that they are also a part of the innate immune system. MRE11 acts as a cytosolic sensor of double-stranded DNA (dsDNA) in association with RAD50 (NBS1 is not involved) and cells lacking MRE11-RAD50 have defects in dsDNA-induced type 1 IFN production [93]. This is because MRE11-RAD50 is required for activation of STING and IRF3. Defects in RAD51 result in an accumulation of host DNA in the cytoplasm, which stimulates the STING pathway, resulting in downstream activation of IFN production [94]. Therefore, it is possible that the recruitment of MRN and RAD51 to the viral genome is another defense against innate immunity, simultaneously preventing MRE11-RAD50 localization to the cytoplasm, and inappropriate processing of host DNA by RAD51 when the MRN complex is disrupted by viral infection. Therefore, in addition to the roles that the viral proteins play in downregulating the IIR, viral DNA replication may control innate immunity by recruiting these DNA damage and repair proteins to the viral genome. In addition, the disruption by E6/E7 of host repair functions would also prevent processing of host DNA that could generate DNA fragments that would otherwise egress to the cytoplasm and activate the IIR [49]. 

Recently we demonstrated that the ISG SAMHD1 (sterile alpha motif and HD-domain containing protein 1) is transcriptionally downregulated by HPV16 [47]. SAMHD1 is at the crossroads of innate immunity and DNA replication/damage [95]. It is often mutated in Aicardi–Goutieres syndrome (AGS), a rare Mendelian disorder that has a constitutively active cGAS-STING pathway and increased interferon production [95]. SAMHD1 is a restriction factor for HIV-1 infection and encodes dNTP hydrolase enzyme activity [96,97,98,99,100,101,102]. In G1, SAMHD1 forms a tetramer that has hydrolase activity to regulate the dNTP pool. Following cell cycle progression into the S phase, SAMHD1 is phosphorylated by CycA-CDK and this attenuates the dNTPase activity, providing increased cellular dNTPs during the S phase [103]. SAMHD1 also functions as an important DDR factor that assists with HR repair of double-stranded DNA breaks [104,105,106,107,108]. It stimulates the 3’ to 5’ exonuclease activity of MRE11 and therefore works at the double strand break with the MRN complex to prevent interferon production [106,108]. As well as regulating HIV1, SAMHD1 regulates the life cycles of several DNA viruses [109,110,111]. What is the role of SAMHD1 during the HPV16 life cycle? CRISPR removal of SAMHD1 results in an aberrant life cycle with enhanced cellular proliferation and viral DNA replication and DNA damage [112]. SAMHD1 is phosphorylated in HPV16-positive cells, switching off the hydrolase activity and potentially promoting its role in DSB repair (James and Morgan, unpublished). Interestingly, the role of SAMHD1 in DNA repair is independent from its hydrolase activity [106]. The role of SAMHD1 in HR is to promote binding of CtIP with the MRN complex, boosting the enzyme activity of MRE11 during HR [106]. Therefore, when we remove SAMHD1 from HPV16-positive cells, the increased DNA damage and cell proliferation is likely due to the aberrant processing of HPV16 DNA during the viral life cycle. We have already demonstrated recruitment of SAMHD1 to the viral genome during replication (James, Das and Morgan, unpublished) and are currently investigating the precise role for SAMHD1 during HPV16 DNA replication. Others have shown that SAMHD1 depletion creates genomic instability and activation of DNA damage signaling which stimulates the innate immune response [107]. Therefore, the interaction between HPV16 and SAMHD1 occurs at a direct crossover between the DDR and the IIR.

Recently we demonstrated that the Werner helicase (WRN) is a member of the HPV16 DNA replication complex [55]. Mutations in WRN result in Werner syndrome; patients with this syndrome have premature aging and a predisposition to cancers [113,114]. CRISPR removal of WRN results in elevated E1-E2 DNA replication levels that are of a poorer quality. As with SAMHD1, WRN is mutated in AGS; the mutation frequencies differ, but the link is intriguing [115]. WRN is a nucleolar protein that enters the nucleoplasm following DNA damage and then interacts with damaged DNA [116]. It is a member of the RecQ family of helicases (operating in a 3’ to 5’ direction) and has a 3’ to 5’ exonuclease activity [117]. WRN is involved in maintaining and repairing DNA replication forks following DNA damage [118,119,120,121]. Providing a further link to HPV replication, the WRN exonuclease domain protects DNA from pathological MRE11/EXO1-dependent degradation [122]. In summary, SAMHD1 and WRN both have direct roles in regulating the MRN complex. Both are involved in regulating the HPV life cycle and it is likely that they both have a critical role in interfacing between DNA replication and damage and suppression of the IIR. Like SAMHD1, depletion of WRN results in the aberrant processing of DNA DSBs and this could result in cytoplasmic accumulation of DNA that would activate the STING pathway to produce interferon [122,123,124]. The disruption of SAMHD1 and WRN function by HPV16 could recruit these proteins to the viral genome and prevent their processing of host DNA damage that could result in the production of cytoplasmic DNA. 

## 5. Conclusions and Future Perspectives

This review summarizes our understanding of the HPV components regulating the DDR and IIR. Several factors during the viral life cycle target both functions to some degree. The regulation of the DDR by E6, E7 and E1 is well established, and E2 contributes by activating viral DNA replication in association with E1. E2 binds to several host proteins involved in the DDR and therefore could contribute directly to DDR regulation. Our recent demonstration that HPV16 genomes activate the DDR independently from E6 and E7 expression demonstrates that viral replication likely contributes to activation of the DDR. Indeed, we could see no difference in the DDR between cells with wild type viral genomes and those that lacked expression of E6 and E7. Overexpression of E6, E7, E2 and E1 can all repress the IIR. However, another striking observation from our recent paper is that disruption of E6 or E7 expression blocks the ability of HPV16 to suppress the IIR. HPV16 genomes with stop codons in E5 also result in activation of the IIR. Importantly, E6 or E7 disruption did not substantially alter the DDR in the HPV16-containing cells. Reactivating the IIR in cells with a persistent DDR could boost the IIR; in HPV16-containing cells, activation of the IIR is enhanced in the absence of E6 or E7 when compared with control cells lacking HPV16. This result opens the therapeutic option of blocking the ability of E6, E7 or E5 (and perhaps E1 and E2) to suppress the IIR. Reactivation of host innate immunity in the infected cells would assist with the adaptive immune response to HPV-infected cells, and potentially boost immunotherapy treatments for HPV cancers. 

## Figures and Tables

**Figure 1 pathogens-09-00467-f001:**
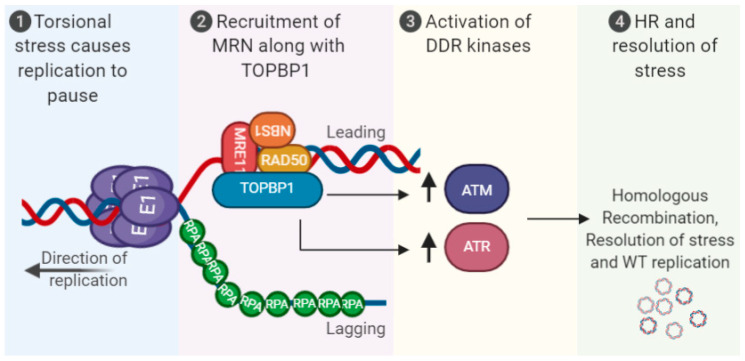
Activation of the DNA damage response (DDR) via human papillomaviruses (HPV) replication stress. Torsional stress on the replicating 8 kbp genome results in replication pausing, recruitment of MRN along with TopBP1 and associated proteins. This promotes activation of the DDR kinases ATR/ATM and homologous recombination (HR) resolution of the stress to promote high-fidelity viral replication. The viral oncogenes E6 and E7 can also induce the DDR, and replication stress, see text for details. MRN (MRE11-RAD50-NBS1).

**Figure 2 pathogens-09-00467-f002:**
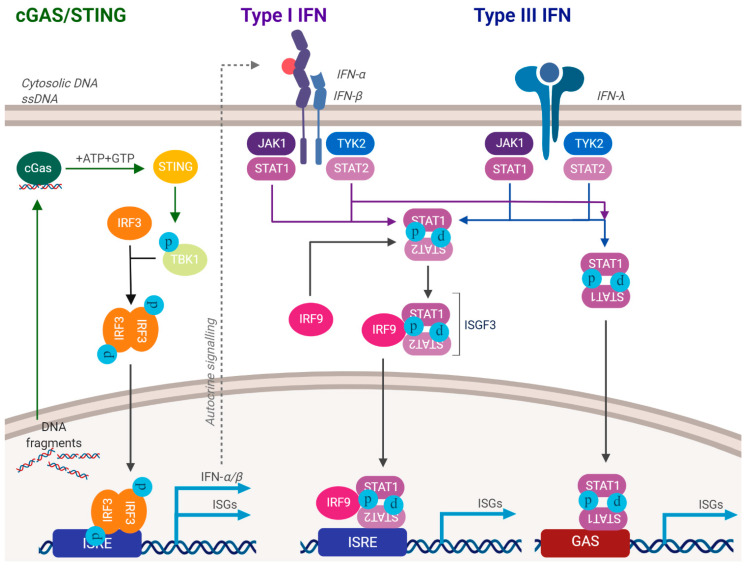
Activation of the innate immune response. DNA damaging agents and/or aberrant release of DNA fragments into the cytoplasm can activate the innate immune response. This can often attenuate cellular growth. As HPV activate the DDR, they must suppress the innate immune response. See text for details.

**Figure 3 pathogens-09-00467-f003:**
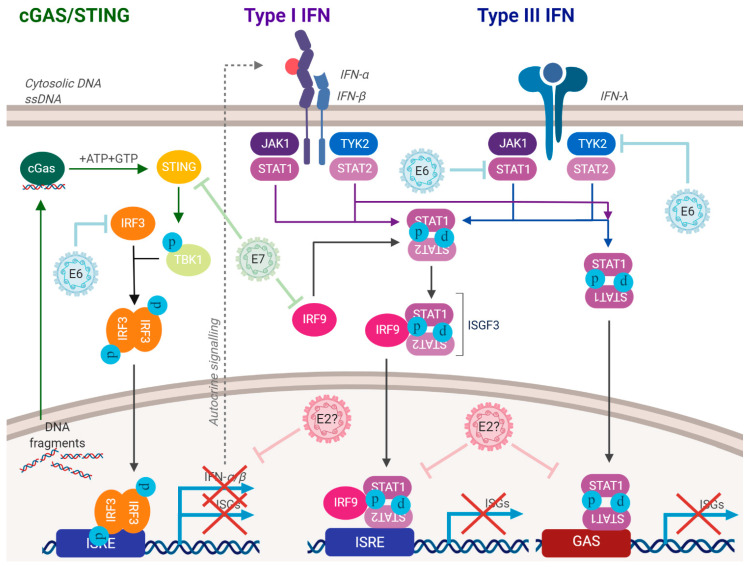
HPV have multiple mechanisms for suppressing the innate immune response. E6 and E7 can repress the innate immune response pathways, some of their mechanisms have been identified. E2 can also repress the innate immune response, but the mechanism is unknown. E5 can also regulate the innate immune response. These processes allow HPV-infected cells to progress through the cell cycle with an activated DDR. This repression also blocks the production of interferon. See text for details.

**Figure 4 pathogens-09-00467-f004:**
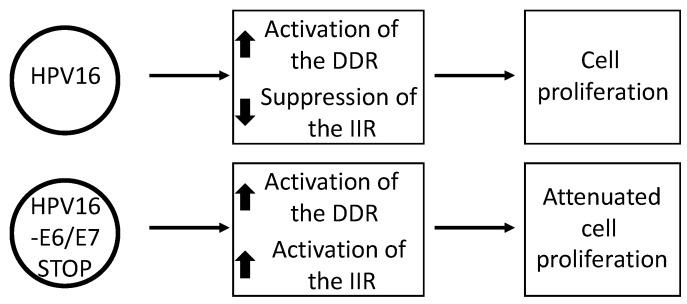
HPV16 E6 and E7 are required for the suppression of the innate immune response (IIR). N/Tert-1 cells (TERT immortalized foreskin keratinocytes) containing an episomal, intact HPV16 genome have an activated DDR and a repressed IIR. However, a stop codon in either E6 or E7 blocks the repression of the IIR and if the expression of both oncogenes is blocked, this significantly attenuates cell growth.

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
