# Peer review of "Activating the DNA Damage Response and Suppressing Innate Immunity: Human Papillomaviruses Walk the Line"

_pathogens, 2020, doi:10.3390/pathogens9060467_

Round 1

Reviewer 1 Report

This review summarizes our knowledge in the HPV field on the manipulation of the DNA repair pathway and the innate immune response in cells infected with high-risk HPV types. It addresses a subject that is of great interest to the scientific community. The authors also describe very nicely how these two signaling pathways are linked and how HPV disturbs these signaling pathways through interaction with central regulators. The work is well-structured, well written and easy to understand and deserves publication in its current form. Only a few minor points need still to be considered:

Line 103: “interacts” needs to be “interact”

Line 104: “viral replicating DNA” need to be “replicating viral DNA”

Line 133 onwards: Second paragraph needs some introductory sentences for better transition from the first to the second paragraph.

Author Response

We thank the reviewer for their positive comments about this manuscript. We have amended as suggested by correcting the spelling/language errors (thank you), and have added a couple of link sentences as suggested.

Reviewer 2 Report

This is a well written and timely review on HR-HPV replication, the double stranded DNA damage response (DDR) process and links to innate immunity. There is some repetition as regards DNA replication and DDR in the review that could be tidied up. For example, it might be best to introduce the MRN complex early on in a brief introduction to DDR rather than having the information in a couple of places in the text.

There are three rudimentary figures, and this aspect of the review could be improved. A figure showing the interplay of HPV proteins interfering with IIR would be helpful and a figure detailing DDR, together with a figure explaining the authors' hypothesis of HPV replication control by DDR, for the non-specialist reader. 

In the paragraph beginning on line 47, numbers of viral genomes quoted are not referenced. This is important because the authors are quite specific about this, yet to my knowledge there have been few accurate analyses of genome copy number at different stages of the HPV life cycle, and a range of numbers has been published by various groups.

In the paragraph commencing line 176 class of IFN is not define, is this why there are spaces in these lines? Mentioning the IFN class is important.

Author Response

We thank the reviewer for their positive comments about the manuscript, it is appreciated. 

We have altered the figures as suggested by the reviewer and this has improved the manuscript, thank you for the suggestion.

We retain our current description of the MRN complex. It needs to be talked about at one point and described at another. We think the current layout will prevent readers having to track back, which can be frustrating in a review.

We agree with the reviewer about the viral genome copy numbers and have amended accordingly. It is clear that in establishment there is a 20-50 copy number as determined with Southern blots in lesions and cell line models. It is clear that amplification in the differentiated tissue occurs in the upper layers, but it has not been precisely quantitated exactly how much amplification occurs. This would require single cell sequencing as not all cells amplify the genome, and to our knowledge this has not been done. 

We have now defined this as class 1 interferon, thank you for raising this important point.